# Typhoon Disaster Risk Assessment Based on Emergy Theory: A Case Study of Zhuhai City, Guangdong Province, China

**Zhicheng Gao [1], Rongjin Wan [1], Qian Ye [2], Weiguo Fan [3], Shihui Guo [2], Sergio Ulgiati [4,5] and Xiaobin Dong [1,6,7,*]**

[1] School of Natural Resources Science and Technology, Faculty of Geographical Science, Beijing Normal University, Beijing 100875, China; gzcecology@163.com (Z.G.); 201821051147@mail.bnu.edu.cn (R.W.)

[2] State Key Lab of Earth Surface Processes and Resource Ecology, Beijing Normal University, Beijing 100875, China; qianye@bnu.edu.cn (Q.Y.); gsh891011@163.com (S.G.)

[3] Department of Economics and Management, North China Electric Power University, Baoding 071003, Hebei, China; fwgcnan@163.com

[4] Department of Science and Technology, Parthenope University, 80133 Napoli, Italy; sergio.ulgiati@uniparthenope.it

[5] School of Environment, Beijing Normal University, Beijing 100875, China

[6] State Key Laboratory of Earth Surface Processes and Resource Ecology, Faculty of Geographical Science, Beijing Normal University, Beijing 100875, China

[7] Joint Center for Global change and China Green Development, Beijing Normal University, Beijing 100875, China

* Correspondence: xbdong@bnu.edu.cn; Tel.: +86-10-58807058

**Abstract:** Typhoons and cyclones are the most impacting and destructive natural disasters in the world. To address the shortcomings of a previous typhoon disaster risk assessment (for example, human factors were involved in determining weights by importance, and this affected the experimental results), an emergy method, which converts energy flows of different properties into the same solar energy basis for a convenient comparison, was used to assess the risk of regional typhoon disasters. Typhoon disaster-related data from 2017 were used to develop an index system including resilience, potential strength, and sensitivity which was in turn applied to assess typhoon disaster risks in Zhuhai City, Guangdong Province, China. The results showed that the spatial distribution of the typhoon disaster risks in Zhuhai significantly differed, with the highest risk in Xiangzhou district, the second highest risk in Doumen district, and the lowest risk in Jinwan district. In addition, improving the level of regional resilience can effectively reduce risks from typhoon disasters. The application of the emergy method in a typhoon disaster risk assessment may provide some theoretical support for national and regional governmental strategies for disaster prevention and reduction.

**Keywords:** emergy method; risk assessment; typhoon disaster

## 1. Introduction

Typhoons are disasters with broad impacts worldwide. The worldwide frequency of severe meteorological disasters and the extent of loss caused by these disasters have increased significantly [1]. Apart from flood disasters, typhoon disasters cause the largest economic losses and casualties in the world [2]. Due to the rapid development of China's industry, warming has been even more severe than other Asian countries. The warming trend has increased greatly, starting in the 20th century at an average rate of 0.03 degrees Celsius per year [3]; studies have shown that global warming

will increase the frequency and intensity of typhoon disasters [4]. At the same time, the frequency of extreme disasters, such as heavy rains, floods, typhoons, and freeze events, has also increased, greatly impacting the Chinese economy, ecosystems, and environment [5]. Compared with other Asian countries, China has had more typhoons with wider impacts causing substantial losses [6]. Due to the monsoon climate, strong summer winds, and rainfall, typhoon disasters along the southeastern coast of China are particularly frequent, and damage has clearly increased as well. Another reason for the substantial losses caused by typhoon disasters is that the southern part of China has a well-developed economy, a dense population, and concentrated social wealth. According to the latest research results, the frequency of severe meteorological disasters and the extent of loss caused by typhoon disasters have increased significantly [7]. During 1982–2006, typhoons that made landfall in China caused 472 deaths and \$4.1 billion in direct economic losses annually; during 2004–2015, the casualties and direct economic losses caused by typhoons that made landfall accounted for 50.2% and 18.3%, respectively, of all meteorological disasters impacts in China [8]. In recent years, in comparison to earlier typhoons, several super typhoons that landed in China have caused greater losses and more serious consequences. For instance, according to data from the Yearbook of Meteorological Disasters in China of 2016, super typhoon "Nepartak" caused 89 deaths and, \$1.8 billion of direct economic losses and affected more than 874,000 people.

A disaster risk assessment is a relatively advanced measurement or technique for disaster prevention, mitigation, and management worldwide [8], and can be considered one of the most important scientific issues today, as well as for the future. It is the basis for a comprehensive disaster reduction and emergency management of typhoon events; nevertheless, there is no uniform assessment standard for this kind of risk [9]. The disaster risk index (DRI), developed by the United Nations Development Program, is representative of disaster risk management at a global scale. The DRI proposes a global scale to a national scale vulnerability assessment indicator system, using death toll, mortality, and mortality relative to the affected population as its risk indicators [10]. Another assessment theory is the multirisk assessment, which combines all natural and technical factors to assess the potential risks in a specific region. This evaluation method has been widely used in Europe [11]. At present, scholars often use the natural disaster risk assessment method proposed by the International Disaster Reduction Agency (IDRA) to evaluate typhoon disaster risk [12]. This method combines hazard, vulnerability, and exposure and then assigns weights to them to obtain the final disaster risk zoning map. Most current research uses three indexes—including hazard, vulnerability, and exposure—to assess the risk of typhoon disasters [4]. Most current research focuses on analyzing hazards [13]. The analytic hierarchy process has been used to determine the weights of parameters such as wind speed, and to then construct a typhoon disaster vulnerability distribution map that allows the typhoon disaster risk to be assessed. Many scholars have used this method to conduct typhoon disaster risk assessments at different scales in China. For example, Lu et al. assessed the risk of typhoon disasters in Zhejiang, China [14], whereas Song et al. conducted a risk analysis at a city scale [15].

However, these studies usually assumed that the indicators were independent of each other and that the weight of each component was determined by the importance of indicators [16]. This approach could lead subjective human factors affecting the experimental results. In addition, the potential relationships between the indicators, including hazard, vulnerability, and exposure, were neglected, and the links between them were separated. Typhoon disaster risk is not easy to accurately assess because human factors will affect the experimental results. In addition, in previous studies, hazard, vulnerability, and exposure indicators were defined in a different way, and thus it is difficult to compare indicators with different properties [17]. For example, for exposure indicators, we can use currency to compare investment and loss in a region, but it is difficult to compare the exposure indicators with the vulnerability indicators. The lack of methods or indicators that can reflect biophysical processes is the main issue in understanding risk disasters. To compensate for the shortcomings of previous research, we use the emergy accounting approach [18], which originated from ecological economics, to assess the risk of typhoon disasters. The emergy method converts different types of available energy into the

same kind of energy (e.g., solar) for easier comparison, thereby quantitatively analyzing the tangible value of natural systems and human socioeconomic systems, resources, and the environment, as well as the relationships between them [19].

On the basis of the emergy results, our study established a systematic typhoon disaster risk assessment framework and analyzed the compositional factors of typhoon disaster risk. Then, the emergy framework was applied to assess the regional typhoon disaster risk. Finally, GIS (geographic information system) was used to visualize the spatial differences in typhoon disaster risks. We applied this emergy framework to Zhuhai City, Guangdong Province, China, to assess typhoon threats to different regions and then to identify areas that are more vulnerable to typhoon disasters. Typhoon is a huge emergy input compared to the average emergy density, which makes it more difficult to measure, yet it provides more scientific and quantitative alternative for policy-makers. This study aims to provide a theoretical reference for disaster prevention and reduction.

## 2. Data and Methodology

### 2.1. Study Site

Zhuhai, located in Guangdong Province in southeastern China, is the core city on the western bank of the Pearl River Estuary (Figure 1). Zhuhai is one of the central cities in the Pearl River Delta and an important node city in the Greater Bay Area of Guangdong, Hong Kong and Macau. The climate in Zhuhai is typical of the southern subtropical monsoon maritime climate with a high annual temperature of approximately 22.5 degrees Celsius from 1979 to 2000. Additionally, Zhuhai has a humid climate with an average annual relative humidity of 80% and abundant rainfall. The annual average rainfall reaches 2061.9 mm. Zhuhai City has a total land area of 1711.24 km$^2$ with three administrative districts, namely, Xiangzhou district, Jinwan district, and Doumen district, including 15 towns and 9 streets. According to statistics, the total population at the end of 2017 was approximately 1.18 million [20]. The disaster events in Zhuhai are mainly heavy rains and typhoons that occur mostly from June to October, with an average of occurring approximately four times per year. Typhoons that seriously affect Zhuhai occur on average once a year, and heavy rain events occur approximately five times a year. Due to abnormal monsoon activity, Zhuhai has become a potential high-risk area for typhoon disasters and is one of the key typhoon-proof cities in the country [21]. Statistics show that Zhuhai's losses caused by typhoon and rain disasters in the past 10 years have exceeded USD 14 billion [22]. Typhoon disasters have a great impact on the economic development of Zhuhai, restricting further development of its economy.

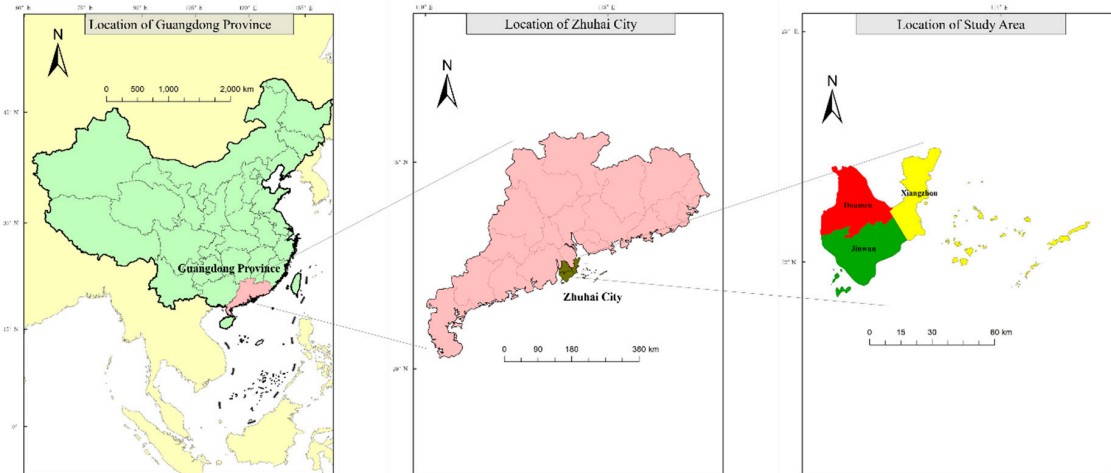

**Figure 1.** Zhuhai City administrative map.

## 2.2. Data Collection

The 2017 typhoon-related data used in this study include the position, intensity, and maximum wind speed of each typhoon every 2 hours collected from the Tropical Cyclone Yearbook 2017 and the China Meteorological Observatory [23]. Zhuhai land use data in 2017 were derived from the Zhuhai Municipal Bureau of Statistics and the "Zhuhai Land Use Master Plan" [24]. Zhuhai land use data were also derived from the Zhuhai Municipal Bureau of Statistics [24] and "Zhuhai Land Use Master Plan". The Zhuhai Statistical Yearbook 2017 was used to obtain the social, economic, and demographic data of each district, including the total population at the end of the year, population density, GDP per capita, and fixed asset investments. From to the literature [25,26], Zhuhai City's 2017 empower density (total emergy usage of a country or city divided by the area of this country or city per unit time) and emergy money ratio (the ratio of the total emergy input to regional GDP in a region in 1 year) data could be obtained. On the basis of the U.S. dollar, we used the USD exchange rate of 29 December, 2017, posted on the Bank of China website [27]. The 2017 disaster loss data for Zhuhai City were obtained from the Zhuhai Municipal Emergency Management Bureau "Zhuhai City Disaster Statistics Yearbook 2017".

## 2.3. Emergy Theoretical Basis for Risk Assessment

### 2.3.1. The Emergy Accounting Method (EMA)

In the 1980s, the famous American ecologist H.T. Odum proposed the concept and the calculation procedure of emergy accounting therein suggesting a new research direction for the quantitative analysis of ecological and economic systems [18]. EMA provides a measure of quality among different forms of energy. It's also provides biophysical perspectives of social and economic systems to complement market-based assessment. It is a measure of the cumulative environmental requirements of resources by a process. It is a unifying metric into which a large number of resource inflows of a process or an economy can be translated and combined in a meaningful way, preserving information on both the quantity and quality of resources and, by doing so, avoiding the need for arbitrary weightings. The emergy approach converts all flows of available energy (exergy) of material and energy resources into units of the same kind of energy (usually of the solar kind). Conversion factors named "transformies" or unit emergy values (UEV) are used for the calculation, expressing the different qualities of each resource flow provided to a system. The rationale here is that resources are generated by the work of the biosphere and therefore emergy expresses the amount of such work to generate a unit of product or service. Depending of the biosphere production processes, not all the materials or energy resources have the same "supply cost", which can be regarded as a resource quality factor because all kinds of energy are directly or indirectly from solar energy. EMA uses solar energy as the benchmark to convert different types of available energy in the system into the same standard solar emergy through solar-based quality factors. The emergy calculation formula is:

$$U = \sum \tau_i \times B_i \tag{1}$$

where U is the solar emergy (sej), $\tau_i$ is the solar transformity (sej/J) or UEV (sej/g), and $B_i$ is the available energy or the mass of the i–th input flow (expressed as J (joule) or g (gram) or other units). Equation (1) originates an inventory table, to list all the input flows and convert them to total emergy and performance indicators [18].

An important part for the emergy approach is the emergy biosphere baseline (GEB) (reference baseline). It represents an estimate of the total emergy annually available to the geobiosphere that drives all environmental processes, serving as the basis for the calculation of resources. In the *Journal of Ecological Modelling*, Brown and Ulgiati [28] updated the GEB (12.0E + 24 sej/year) based on earlier methods proposed by Odum [18] and refinements by Brown and Ulgiati [29] (9.44E + 24 sej/year). The emergy calculations involved in this study were multiplied by a factor of 1.28 (ratio of new baseline to old baseline) to be consistent with the latest GEB [28].

Emergy analysis theory has been gradually improved in recent years and has been applied successfully to various sectors and fields. The basic steps of the common energy value analysis are as follows: (1) collect information on the natural environment, social resources, and economic activities for the countries in which the study is being conducted or the country in which the city is located; (2) system diagram—defining the system's outer boundary and the intersystem components of the investigated system using graphic symbols [18] to express components, processes, and energy flows related to the system; (3) establishing an emergy accounting table (converting all flows into emergy units in order to be added into a total emergy used (U)); (4) calculating performance indicators relating resource use, products, and economic value; (5) establishing a suitable emergy analysis indicator system with reference to the research objectives and purposefully analyze the regionally related research content; and (6) analyzing the results according to the set of calculated indicators and providing policy suggestions.

Combining the above emergy analysis steps with disaster risk theory, the framework of this study is as follows (Figure 2): (1) Inventory—collecting typhoon data, socio-economic data, and emergy data of Zhuhai City (Section 2.2.). (2) System diagram—analysis of the typhoon energy flow process in nature–agricultural system and urban system based on defining the system boundary (Figure 3). (3) Indicator system—establishing an indicator system for typhoon disaster risk assessment including total driving emergy, sensitivity index, typhoon strength effectiveness, resilience, and typhoon risk comprehensive index (Table 1). (4) Indicator calculation—calculating the emergy of various indicators in each district of Zhuhai City (Table 2). (5) Risk maps—drawing indicators' risk maps of typhoon disaster in Zhuhai City according to the calculation results of various indicators. (6) Results—analyzing and explaining the risk situation of each district in Zhuhai City according to the calculation results and risk maps of various indicators.

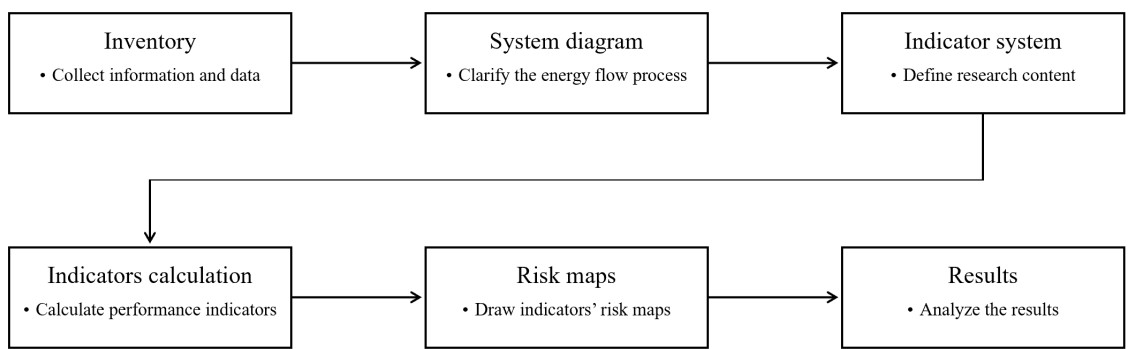

**Figure 2.** Flow chart of typhoon disaster risk emergy analysis.

**Table 1.** Emergy indicators of typhoon disaster risk (indicator system).

| Indicators | Interpretation | Unit |
| --- | --- | --- |
| Total driving emergy | Typhoon energy × Solar transformity | sej |
| Sensitivity index | (Different land use area × empower density)/(unused land area × empower density) [1] | - |
| Typhoon strength effectiveness | Total driving emergy × Sensitivity index | sej |
| Resilience | Medical staff per 10,000 people × Solar transformity | sej |
|  | College students per 10,000 people × Solar transformity | sej |
|  | Shelterbelt area × Solar transformity | sej |
|  | Foundation protection weight × Solar transformity | sej |
|  | GDP × Solar transformity | sej |
|  | Fixed asset investment × Solar transformity | sej |
| Typhoon risk comprehensive index | Typhoon strength effectiveness/Resilience | - |

[1] Including cultivated land, forestland, residential land, traffic land, and unused land. Empower density refers to the total emergy usage of a country or city divided by the area of this country or city per unit time.

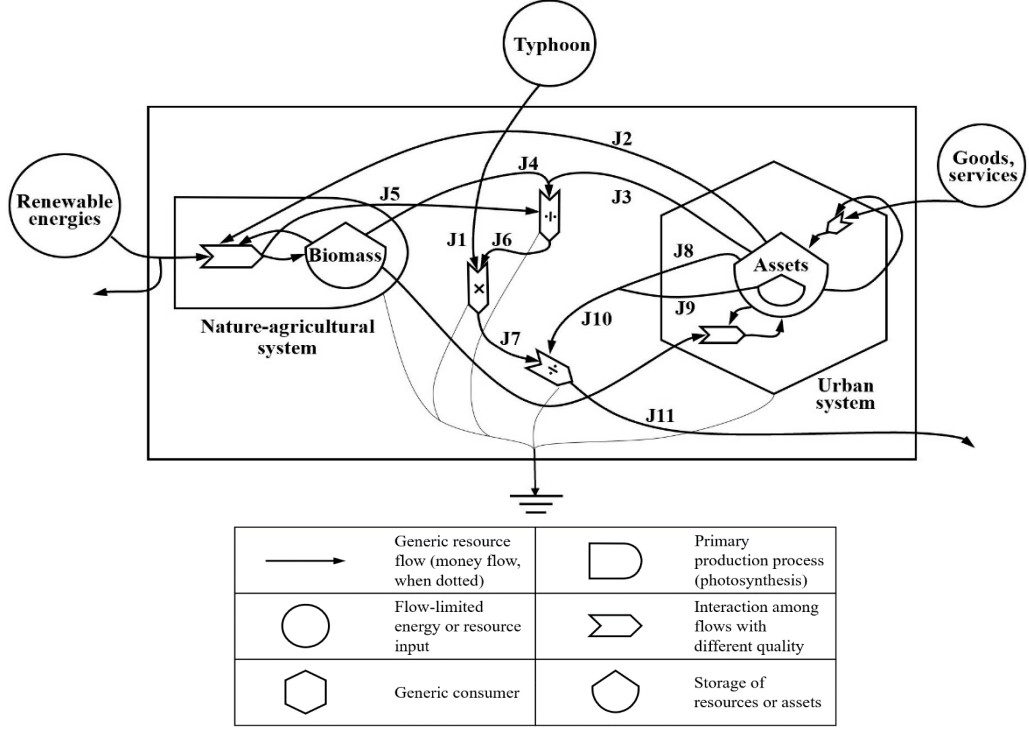

**Figure 3.** Typhoon disaster energy flow system. Symbols are from Odum [18].

### 2.3.2. Typhoon Disaster Energy System Diagram

A typhoon disaster energy system diagram consisting of natural systems, agricultural systems, and urban systems is the basis for analyzing the complex relationship between the various components of typhoon disaster risk [18]. Figure 3 illustrates the linkages and differences between energy and material flows in ecosystems and economic systems. When a typhoon occurs, the typhoon energy representing the total driving emergy (J1) passing through a certain area will put the area at risk. Agricultural system sensitivity (J2) and urban system sensitivity (J3) represent the energy lost by the agricultural system and urban system, respectively, when extreme weather events occur, and the total energy lost by the two is represented by agriculture–urban system sensitivity (J4). Natural system sensitivity (J5) is expressed in terms of unused land energy. The sensitivity index (J6) represents the ratio of the energy stored in the system to the energy of the unused land. The higher the ratio is, the greater the loss in the event of a typhoon. Extreme weather events such as typhoons have an impact on the surrounding environment. Environmental changes affect the area to a certain extent, expressed as the typhoon strength effectiveness (J7), which is related to the total driving emergy (J1) and the sensitivity index (J6). Factors affecting typhoon disaster risk include natural factors, population factors (J8), and socioeconomic factors (J9), which together reflect the system's resistance (J10) and express the region's ability to respond to disasters. Using the resilience and typhoon strength effectiveness, the typhoon risk comprehensive index (J11) was obtained to assess the risk of regional typhoon disasters.

### 2.3.3. Typhoon Disaster Risk Assessment Model

As mentioned earlier, regional typhoon disaster risks are highly correlated with typhoon total driving emergy, sensitivity index, impacted area resilience, and historical disaster losses. By analyzing the compositional factors of typhoon disaster risk and understanding the relationship between various components, we constructed an emergy indicator system to assess regional typhoon disaster risk. Table 1 shows how these emergy indicators of typhoon disaster risk can be defined and constructed.

### 2.3.4. Typhoon Total Driving Emergy

The total driving emergy represents the total cumulative strength of typhoons that hit an area during a period (1 year). It refers to emergy carried by the hazard when it occurs. Using a typhoon as an example, the total driving emergy of a typhoon is equal to typhoon energy times solar transformity. From Odum's study [18] and the new reference baseline (12.0E + 24 sej/year) [28], we also determined that the solar transformity for typhoons is approximately 8414 sej/J (6573 sej/J × 1.28). The calculation process is as follows:

Transformity of typhoons ($\tau$) = Previous transformity [18] × (New baseline [28]/Old baseline [18])

= 6573 sej/J× (12.0E + 24 sej/year/9.3E+ 24 sej/year)

= 8414 sej/J

The following is the total driving emergy formula:

$$U_T = E_T \times \tau \tag{2}$$

where $U_T$ is the total driving emergy (sej), $\tau$ is its solar transformity (sej/J), and $E_T$ is the typhoon energy (J). The largest the emergy of the typhoon compared to the average empower density of environmental and human-made assets of the area, the larger the impact (Figure 3).

In this study, we estimated the typhoon energy by the duration time of typhoons occurrence within the category 7 wind circle (typhoon bottom center maximum average wind speed is greater than 17.1 m/s) in each region in 2017 and the power of the category 7 wind circle [18,29]. The following is the typhoon energy formula:

$$E_T = t \times P_T \tag{3}$$

where $E_T$ is the typhoon energy (J), t is the time in which category 7 wind circle passed through the area (hour), and $P_T$ is the power of the category 7 wind circle (J/year).

### 2.3.5. Sensitivity Index

A sensitivity index refers to the extent to which a system is affected by extreme weather events. Odum and Brown believed that the more intensive the economic activity, the higher the empower density of the land would be [29]. Different types of land use have different energy levels and are affected by typhoons to different degrees. Therefore, a sensitivity index analysis was conducted according to the empower density of the land. For example, commercial and residential land in urban areas often has higher empower density [30]. Some scholars use the ratio of urban area to unused land area to represent the economic development potential of the region [31]. However, according to emergy theory, compared with GDP, emergy can better reflect the regional economic level, that is, the total amount of emergy applied each year is the real wealth of the region. Therefore, we speculate that the ratio of the total accumulation of one different land use area emergy to unused land emergy within a unit time can better reflect the loss of real wealth that may be caused by a disaster in the area. On this basis, the sensitivity index of a district is represented by the ratio of the energy contained in different land use types to the emergy of unused land. The following is the sensitivity index formula:

$$M = \left[\sum (\rho \times A)\right] / (\rho \times A_{nu}) \tag{4}$$

where M is the sensitivity index, $\rho$ is the empower density of different land uses (sej/m$^2$), A is the area of different land uses (m$^2$), and $A_{nu}$ is the unused land area (m$^2$).

### 2.3.6. Typhoon Strength Effectiveness

The typhoon strength effectiveness shows how a system might be affected when an extreme weather event occurs. It suggests that higher strength impacting an extremely sensitive area provide more of an effect. A higher typhoon strength effectiveness will increase the vulnerability of typhoon

disasters. The typhoon strength effectiveness is related to the total driving emergy and sensitivity index of the affected area. The following is the typhoon strength effectiveness formula:

$$I = U \times M \tag{5}$$

where I is the typhoon strength effectiveness (sej), M is the sensitivity index, andU is the total driving emergy (sej).

### 2.3.7. Resilience

Resilience refers to the systematic process of continuous learning and adjustment under a changing environment, which incorporates beneficial events and reduces disaster losses. Resilience is specific to the environment, and the adaptability of different regions and groups varies. For example, the adaptability of urban areas is higher than that of rural areas. Resilience within the region is causally related to regional economic and social conditions. The number of medical staff can effectively reflect the sanitary level of a region, which means that when a disaster occurs, the more medical staff there are, the faster the treatment speed and the ability of the area to recover quickly. The number of students in the region can reflect the overall quality of the regional population to some extent [21]. Some studies suggest that higher population quality can effectively reduce disaster risk in the region [32]. Fixed asset investment represents a region's fixed asset investment situation. In general, the higher the level of investment in fixed assets in a region, the faster the reconstruction after a disaster. The GDP per capita indicates the level of economic benefits in the region and is positively related to adaptive capacity. In many studies using Chinese cities as research areas, the first–level indicators of resilience include demographic factors, economic factors, and social factors [15,31]. The indicators selected in this study are the results of refining various population, economic, and social indicators on the basis of principal component analysis. On the basis of the information above, in this study, we used the number of medical staff per 10,000 people, the number of college students per 10,000 people, fixed asset investment, and GDP to characterize the region's resilience.

**Table 2.** Calculated results of the emergy indicators for typhoon disaster risks in different districts of Zhuhai City (indicators calculation).

| Item [1] | District | | |
|---|---|---|---|
| | Xiangzhou | Doumen | Jinwan |
| Typhoon transit time/hour | 35 | 30 | 31 |
| **Total driving emergy [2]/$10^{22}$sej** | 1.56 | 1.33 | 1.38 |
| Cultivated land area/$10^7 m^2$ | 0.53 | 9.47 | 7.60 |
| Emergy of cultivated land/$10^{21}$ sej | 0.65 | 11.64 | 9.35 |
| Forestland land area/$10^7 m^2$ | 15.93 | 10.67 | 11.00 |
| Emergy of forestland/$10^{21}$ sej | 9.63 | 6.44 | 6.64 |
| Residential land area/$10^7 m^2$ | 17.47 | 11.20 | 12.60 |
| Emergy of residential land/$10^{25}$ sej | 10.12 | 6.49 | 7.30 |
| Traffic land area/$10^7 m^2$ | 1.66 | 1.60 | 2.13 |
| Emergy of traffic land/$10^{24}$ sej | 4.73 | 4.54 | 6.05 |
| Unused land area/$10^7 m^2$ | 11.73 | 23.13 | 18.33 |
| Emergy of unused land/$10^{20}$ sej | 1.23 | 2.43 | 1.93 |
| **Sensitivity index [3]/$10^5$** | 8.40 | 4.15 | 4.32 |
| **Typhoon strength effectiveness [4]/$10^{27}$sej** | 13.11 | 5.55 | 5.97 |
| Medical staff per 10,000 people/person | 52.08 | 36.22 | 25.71 |
| Emergy/$10^{21}$ sej | 0.70 | 0.49 | 0.34 |
| College students per 10,000 people/person | 1642.01 | 890.94 | 369.89 |
| Emergy/$10^{21}$ sej | 3.84 | 2.08 | 0.87 |
| Fixed asset investment/billion USD | 100.02 | 21.55 | 44.63 |
| Emergy/$10^{22}$ sej | 6.42 | 1.38 | 2.87 |
| GDP/billion USD | 175.64 | 35.60 | 56.27 |
| Emergy/$10^{22}$ sej | 11.28 | 2.29 | 3.61 |

**Table 2.** *Cont.*

| Item [1] | District | | |
|---|---|---|---|
| | Xiangzhou | Doumen | Jinwan |
| **Resilience emergy [5]/$10^{22}$ sej** | 3.95 | 1.91 | 3.40 |
| Agricultural emergy loss/$10^{21}$ sej | 4.35 | 1.49 | 1.24 |
| Mining and business emergy loss/$10^{21}$ sej | 2.37 | 0.43 | 0.61 |
| Infrastructure emergy loss/$10^{19}$ sej | 306.14 | 0.19 | 8.47 |
| Public facility emergy loss/$10^{19}$ sej | 89.33 | 0.19 | 8.86 |
| Property emergy loss/$10^{20}$ sej | 10.26 | 0.51 | 0.73 |
| **Direct economic emergy loss /$10^{21}$ sej** | 5.51 | 4.28 | 3.60 |
| **Typhoon risk comprehensive index [6]/$10^{5}$** | 3.31 | 2.90 | 1.75 |

[1] Energy data and transformity (e.g. typhoon, students, and medical staff) are a synthesis of data from Zhai et al. [26], Odum [18], Brown and Ulgiati [33], Ghisellini et al. [34], and Brown and Ulgiati [28]. Cultivated land empower density = 1.2301E + 14 sej/(m$^2$ · a) [25]. Forestland land empower density = 6.044E + 13 sej/ (m$^2$ · a) [25]. Residential land empower density = 5.79593E + 17 sej/(m2 · a) [25]. Traffic land empower density = 2.83673E + 17 sej/(m2 · a) [25]. Unused land empower density = 1.05E + 12 sej/(m2 · a) [25]. All emergy money ratio data came from The Zhuhai Statistical Yearbook 2017 (emergy money ratio of fixed asset investment = 6.41961E + 19 sej/ USD, emergy money ratio of GDP = 6.40751E+17 sej/ USD). [2] Total driving emergy = Typhoon transit time × Typhoon consumes energy per hour [33] × Transfromity (8414sej/J). [3] Sensitivity index = (Emergy of cultivated land + Emergy of forestland + Emergy of residential land + Emergy of traffic land + Emergy of unused land) / Emergy of unused land. [4] Typhoon strength effectiveness = Total driving emergy × Sensitivity index. [5] Resilience emergy = Emergy of medical staff per 10,000 people + Emergy of college students per 10,000 people + Emergy of fixed asset investment + Emergy of GDP. [6] Typhoon risk comprehensive index = Typhoon strength effectiveness / Resilience emergy.

### 2.3.8. Typhoon Risk Comprehensive Index

The typhoon risk comprehensive index is the ratio of the typhoon strength effectiveness to resilience. It reflects the damage to the system from an extreme weather event. Typhoon risk is positively correlated with the typhoon risk comprehensive index and negatively correlated with resilience. It means the final risk depends on the size (emergy) of the impacting force and the size (again, emergy) of the recovering ability. The following is the typhoon risk comprehensive index formula:

$$V = I/R \tag{6}$$

where V is the typhoon risk comprehensive index, I is the typhoon strength effectiveness (sej), and R is resilience.

## 3. Results

We calculated the total driving emergy, sensitivity index, typhoon strength effectiveness, resilience emergy, and typhoon risk comprehensive index of Zhuhai City in 2017 according to the method described in Section 2.3. The calculation results are shown in Table 2.

According to the results of the data analysis, the following results, and conclusions can be drawn:

1. The total driving emergy represented the total cumulative emergy during 2017. The more energy a region stores over a period, the larger the total driving emergy. Table 2 and Figure 5 show that among the three districts of Zhuhai, Xiangzhou district had the largest total driving emergy value, and Doumen district had the smallest. This was because the typhoon centers of the three typhoons that passed through Zhuhai in 2017 moved from the southeast to the northwest. The Xiangzhou district was closest to the typhoon center, and the Doumen district was the farthest away from the center. In addition, when the typhoon landed, due to the influence of the topography of the underlying surface, the wind force of the typhoon weakened, and thus the total driving emergy value in the Doumen area was the smallest.

2. The concept of energy level is used to analyze the sensitivity of the research area to typhoons. On the basis of the empower density of land use, we calculated the emergy content of different land use types in each district of Zhuhai. Areas with a high ratio of total emergy to unused land emergy (such as residential areas and commercial areas) have high sensitivity, and they were likely

to suffer greater losses from typhoon disasters. The calculation results showed that compared with that of the other two districts, the sensitivity of Xiangzhou district was extremely high. This was because Xiangzhou district has a large population and is also a place where residential and commercial areas are very concentrated. In contrast, this area had a high sensitivity index because it has a minimal amount of unused land

3.  The purpose of the adaptive capacity assessment was to understand the ability of the affected area to respond to typhoons. Xiangzhou district has a developed economy, medical facilities, and a highly educated population, and thus its resilience value was found to be high. In comparison to the other districts, Doumen district has more arable land, the ecosystem is more fragile, and its economic level is relatively undeveloped, so its resilience value was found to be low.

4.  The typhoon strength effectiveness was related to the total driving emergy and sensitivity, which indicated that the system may suffer when an extreme weather event occurs. Due to the dense population of Xiangzhou district, rapid economic development and accumulated assets, the typhoon strength effectiveness was higher there than the other areas when typhoons occurred. In contrast, areas with relatively slow population development, such as in Doumen district, had a lower total driving emergy index.

5.  The typhoon risk comprehensive index was related to the typhoon strength effectiveness and resilience. The greater the typhoon risk comprehensive index in a certain area is, the higher the risk of typhoon disaster. For this index, Figure 4 shows that Xiangzhou district had the highest value, Doumen district had the second highest value, and Jinwan district had the lowest value, which was significantly lower than the first two values. The resilience of Xiangzhou district was the highest among the three districts, but because of the high concentration of population and wealth, the value of the typhoon strength effectiveness was also much higher than that of the two districts; thus, the value of the typhoon risk comprehensive index was the largest. In addition, although the value of the total driving emergy index of the Doumen district was low, its resilience was also the weakest of the three areas, and thus it has a higher risk. As shown in Table 2, the emergy of the direct economic loss due to the typhoon disasters in the three regions is consistent with the results of the typhoon risk comprehensive index obtained in this study.

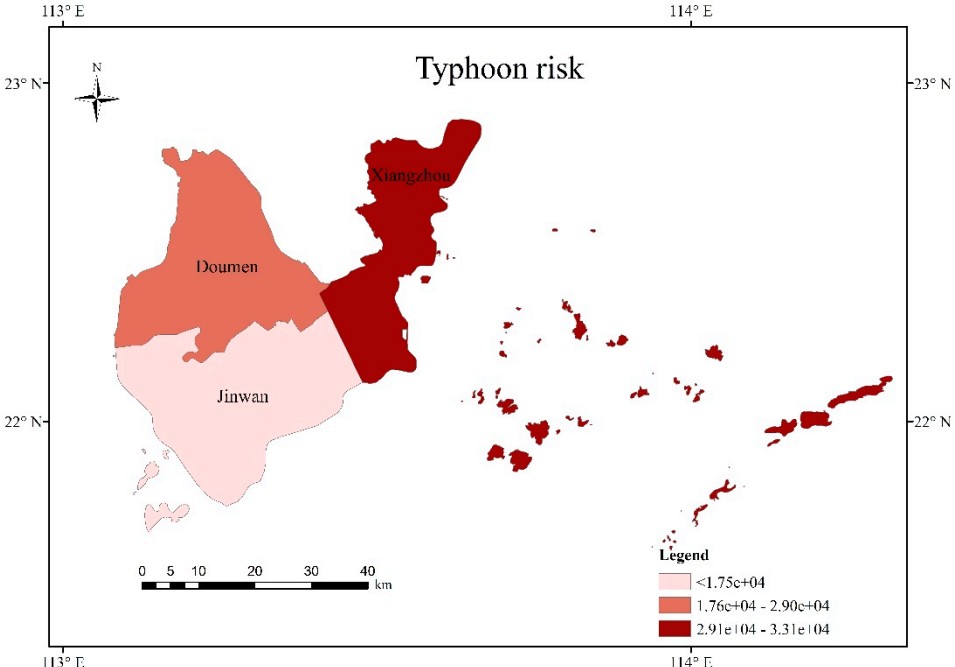

**Figure 4.** Evaluation results of typhoon risk.

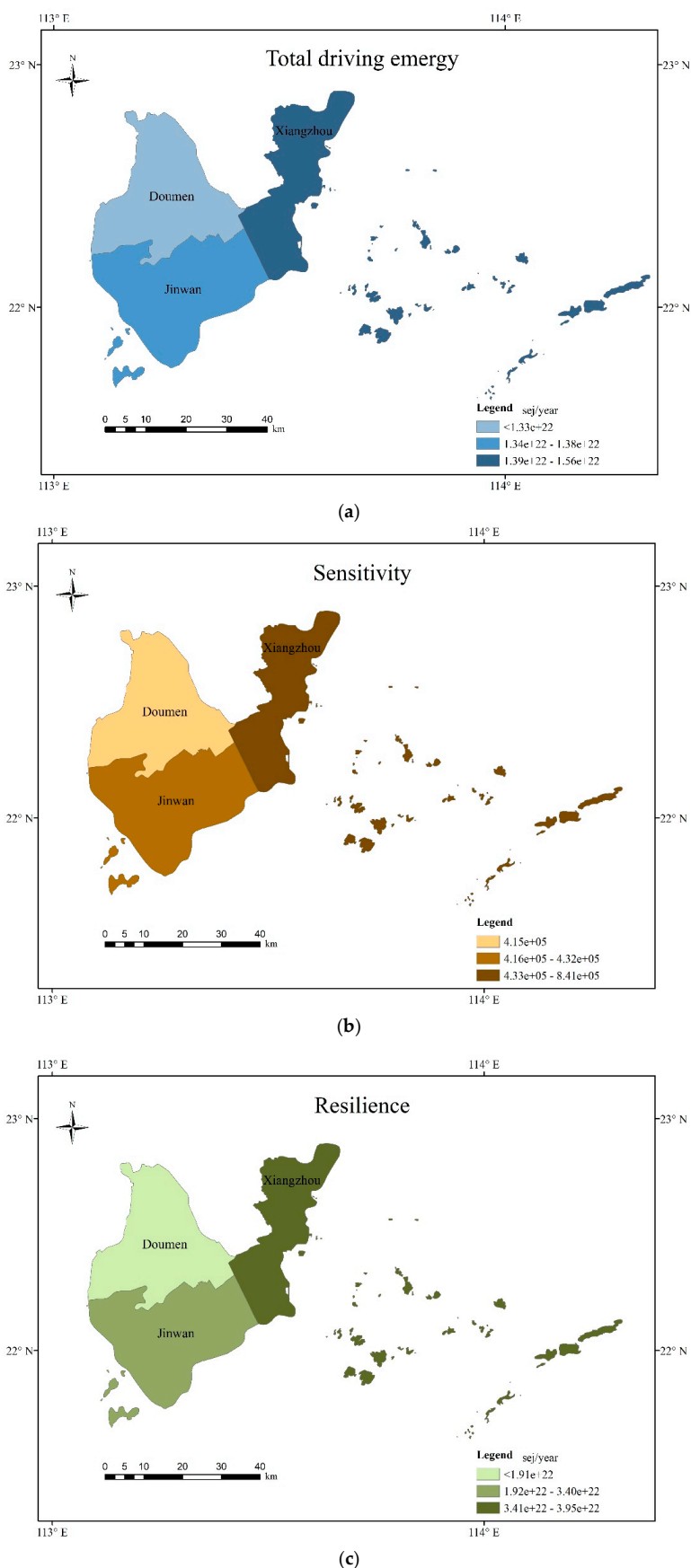

**Figure 5.** (**a**) Assessment results for total driving emergy; (**b**) assessment results for sensitivity; (**c**) assessment results for resilience.

## 4. Discussion

From an ecological perspective, the use of unified emergy units to assess the risk of typhoon disasters provides a new approach and a comparable basis for disaster risk research in different regions. The use of emergy theory in the assessment of the risk from typhoon disasters is an approach worth exploring. The emergy method provides a new concept—unity energy of different properties for calculations and comparisons not only eliminates the factors that cannot be compared between different energy types but also avoids inclusion of human factors (such as the calculation of weights) affecting the experimental results. To make the experimental results more robust, we calculated the losses caused by the typhoon disaster in Zhuhai City in 2017. The calculated results showed that the amount and trend of the losses were consistent with the conclusions of our study. We believe that the emergy methodology used in the disaster risk assessment for Zhuhai provides an alternative reference for other regions. However, there are still some limitations for practical application. For example, due to economic and social development, meteorology, hydrology, and the natural environment constantly evolving over time, the risk indicators of typhoon disasters will change. Only the typhoon disaster risks in 2017 were assessed in this research; if long-term temporal and spatial data are analyzed, then the results would be more robust.

At present, the mostly used typhoon disaster risk assessment methods are the mathematical statistics method based on historical disaster conditions, weighted index method based on indicator system, visualization method based on scenario simulation, and interpretation method based on multi-source remote sensing images. Mathematical statistics method generally evaluates the losses caused by disasters of the same level in the future period on the basis of the historical disaster situation. Because the occurrence of disasters has uncertainty in time and space, even the losses caused by disasters of the same level are not the same, and thus the prediction results of this method are not accurate enough, which was shown in a previous study [35]. The weighted index method based on the indicator system evaluates the hazards, the exposure and the vulnerability of disasters separately, and then combines the evaluation results of the three aspects. For instance, some scholars analyzed the evolution of typhoon disasters characteristics and non-structural disaster avoidance measures in the China coastal main functional area [36]. The disadvantage of this type of method is that the selection of weights is subjective, and different models may have different results. The emergy method used in this study overcomes this shortcoming because the ecosystem can be represented by the quantity and quality of the energy and matter converging in the same system, measured in terms of emergy. Therefore, emergy enables us to identify, quantify, and weigh the inputs that feed the system. The visualization method based on scenario simulation and the interpretation method based on multi-source remote sensing images focus on assessing and simulating disaster losses and future losses [37,38]. The disadvantage of these two types of methods is the higher requirement for data accuracy, besides the face that the subjective evaluation model is still used in these two methods. It is more accurate to use emergy to assess the losses, because emergy can calculate the losses of real wealth in a region rather than the monetary value. In summary, emergy methods can be important and necessary to supplement under current assessment methods.

The indicator system is the core of this assessment. Our indicator system combines emergy theory, the traditional disaster risk assessment indicator system currently used in China, and the international meteorological disaster risk assessment indicator system. Hazard refers to the possibility of a potential disaster occurring at a specific time in an area [32]. The three basic indicators for measuring hazard are: the annual average number of typhoon disasters in this area, the number of deaths caused by typhoon disasters, and economic losses. The total driving energy in this study largely reflected the hazard part in disaster risk assessment. Exposure indicators, of which population density and GDP per capita to show the impact of typhoon disasters on human social systems, could be replaced to some extent by sensitivity and resistance indicators in this study. In addition to these, the emergy disaster losses and GDP emergy can well reflect vulnerability. Overall, our indicator system adds the advantages of emergy theory while retaining key factors of traditional typhoon disaster risk assessment theory.

This system not only avoids the subjectivity of artificial empowerment to a large extent, but also makes the evaluation results have certain practical value.

We believe that emergy methods have broad potential in the field of natural disaster risk assessment, and even in disaster risk science. In terms of natural disasters, every disaster is inseparable from the release of energy from the hazard itself. Because the emergy method is based on the study of the energy flow process, the emergy method is theoretically very suitable for natural disaster research. According to the quality and quantity of data used in disaster risk assessment, the assessment can be divided into a quantitative assessment, semiquantitative assessment, and qualitative assessment. The current research focuses on semiquantitative research, which requires expert indexing or an analytic hierarchy process to determine the weight of each indicator. The advantage of the unified dimension of the emergy method provides theoretical support for a quantitative assessment. In terms of accounting, the emergy method still has advantages. For example, emergy reflects the true value of matter (energy contained throughout an entire life cycle) rather than simple monetary value. Therefore, the emergy method also provides new concepts for calculating indirect disaster losses, which is a popular topic in disaster risk science.

## 5. Conclusions

Comprehensive risk assessment of the disaster has been challenging work because it involves many complex issues. We set up a special indicator system of comprehensive risk assessment of typhoon disasters that was based on emergy theory, which has a common unit that allows all resources to be compared on a fair basis, providing a more comprehensive choice of environmental decision-making. These indicators such as total driving emergy, sensitivity index, typhoon strength effectiveness, resilience emergy, and typhoon disaster risk comprehensive index have been constructed to quantitatively analyze the risks of typhoon disasters on natural and human economic and social systems.

Guangdong Province suffers more typhoon disaster in China. On the basis of typhoon disaster risk assessment indicators established by this research, our results showed that Xiangzhou district is economically developed, has accumulated assets, and has a high total driving emergy index, but because of its high resilience, the evaluation results were found to be slightly lower than expected. Correspondingly, areas with a relatively low total driving emergy index, such as Doumen district, have a large amount of arable land, a fragile ecological environment, and poor self-recovery capabilities, and thus their resilience values are low, and the risk of typhoon disasters remains high. Areas with a low potential intensity index and strong resilience, such as Jinwan district, have a lower risk of typhoon disasters. It was found that improving the resilience of a system can effectively reduce the risks from typhoon disasters.

**Author Contributions:** Conceptualization, Z.G., Q.Y., R.W., W.F., and X.D.; methodology, X.D. and S.U.; software, Z.G. and R.W.; validation, S.G., Q.Y., and R.W.; formal analysis, Z.G.; investigation, Z.G. and R.W.; resources, X.D.; data curation, Z.G.; writing—original draft preparation, Z.G.; writing—review and editing, X.D., and S.U.; visualization, Z.G.; supervision, X.D.; project administration, X.D. and Q.Y.; funding acquisition, X.D. All authors have read and agreed to the published version of the manuscript.

**Funding:** This research was funded by China Science and Technology Supporting Program (2017YFE0100400), the Second Tibetan Plateau Scientific Expedition and Research Program (2019QZKK0608).

**Conflicts of Interest:** The authors declare no conflict of interest.

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
