# Peer review of "Typhoon Disaster Risk Assessment Based on Emergy Theory: A Case Study of Zhuhai City, Guangdong Province, China"

_sustainability, doi:10.3390/su12104212_

Round 1
Reviewer 1 Report
The manuscript provides a good piece of work by applying the Emergy measures to the risk assessment of natural phenomena (i.e., typhoon). The Emergy method is a powerful statistical analysis which give a quantitative measures of the phenomena. Nevertheless, the article has some flaws;
Please unify the citation style (i.e., numbered style), please check lines (165-167 and 282-283)
It may be better to add a definition of each expression as some readers may not be familiar with all expression (i.e., empower density)
Authors adopt the factor 1.28 (see line 169) to be consistent with the latest baseline. Please explain what is the latest baseline? by reference!
In section 2.3.4 Typhoon Total Driving Emergy;
- Please explain how to calculate the total emergy using the time spent and consumed?
- Authors neglected some important factors (e.g., distance to typhoon center, air density and wind radius), please justify!
In section 2.3.7 Resilience;
- Authors used few measures for resilience calculation, please check (Sajjad, M., & Chan, J. C. (2019). Risk assessment for the sustainability of coastal communities: A preliminary study. Science of The Total Environment, 671, 339-350.), for more information and justify why focusing on these factors only.
In Table (2); please explain who to calculate the Sensitivity Index, Resilience emergy and the Direct economic emergy loss? Is it average!
Add appendix for the data used to produce Table 2 (e.g., empower density and emergy money ratio).
Author Response
Thank you for your constructive comments and suggestions. The attachment is a point-to-point reply.

Reviewer 2 Report
In this paper the authors used an unconventional method, i.e. “Emergy method” for risk assessment of natural disasters like typhoon. The authors developed arithmetical relationship based on their understanding on events related to typhoon and quantified different indicators of disaster risks.
Overall, the paper is well written. However, some issues need to be clarified to make the manuscript more readable. The issues raised are mentioned in the appended manuscript.

Author Response

(The authors gave the same response as above.)

Reviewer 3 Report
- are you sure that "Zhuhai city" is a good choice for keyword? It is vague.
- Is reference [1] the most recent one you can provide? Because we are in 2020, but the reference (from 2015) talks about the next 20 years. - Line 39-40: "Due to global warming, China's warming has been even more severe than typhoons." - Please explain the meaning of this sentence, it does not make sense for me. IF there is a mistake, fix it. - Line 69: include acronym for International Disaster Reduction Agency - Line 72: include "on" in the sentence "Most current research focuses ON analyzing hazards [13]" - Line 80: include "to" in the sentence "...approach could lead to subjective human factors affecting..." - Is your study the first one using the energy method to assess impact of typhoon disaster?Figure 1: the legend of this figure does not make any sense to me. Can you explain? If there is a mistake, can you fix it?
-Line 125: Save the references to the end. Refer the website of the China Meteorological Observatory in the references section. The same for Line 127/128 (land use info), and 134 (Bank of China) - Lines 131/132: why are you providing information such as number of doctors per 10,000 people and number of college students per 10,000 people? How does that relate to your work? Make it clear in your text. - Line 132/133: explain what is empower density and energy money ratio, and how you obtained those parameters from the references 22,23 - Line 142: do not capitalize "It" - Line 162: change to "It is also the basis for calculator..." - Line 162: change to "... which is the unit of energy investments from nature...." - Why is section 2.3.1 in methods? You could have a section prior to methods for fundamental background in which you could include the description of the EMA - Lines 164/170 shouldn't be in methods. This is literature review. - Line 188: sometimes you capitalize variables (such as Unit Energy Values in Line 149), and sometimes you do not (such as total driving energy). Pick a standard and be consistent. - Something looks weird in Figure 2. There are some lines that look like they were drawn by hand (at the bottom of the picture). What do they mean? There is no legend for them. They do no look great. - I am not sure what is the necessity of the column units in Table 1, all of them are same units. Just write it at the legend. - Line 265: change 10000 to 10,000 - Line 265/266: after more than one hundred lines, you are explaining lines 132/133. Connect these two thoughts. - Discuss in which level your results are different than the ones presented in [17], [18], and [19]. - Literature review in this manuscript needs to be improved. I am sure there are a lot more works in literature to describe, and compare your dataAuthor Response
Thank you for your constructive comments and suggestions. The attachment is a point-to-point reply.

Reviewer 4 Report
The article is interesting and the subject deserves investigation. However, the manuscript has important shortcomings that need to be corrected both in structure and content. In addition, the text is sometimes somewhat confusing and a clear guiding thread is missing for the research and its objectives. In my opinion, it is not currently in a position to be published. Accordingly, I recommend an in-depth review. If the authors manage to incorporate the necessary improvements in a satisfactory way, that evaluation may be reconsidered. Below, I detail the issues that must be addressed by the authors.
AREA OF STUDY
More information about the study area is missing. The map in Figure 1 is not contextualized across China as a whole, making it difficult to understand the location of the regions for non-Chinese readers. On the other hand, it is necessary to include all the references to cities or places mentioned in the manuscript in this map.
METHODOLOGY AND RESULTS
The methodological approach is somewhat confusing. The authors describe a couple of models, and then enumerate (with a short explanatory text) a list of indicators in the methodology section. However, there is no clear methodological general framework with different stages and a well-defined results target. Furthermore, there is no clear correspondence between what is explained in the methodology and what is presented in the results section. The authors should include at the end of the methodology section a final subsection that clearly summarizes the framework in a schematic way, with an organization that can be clearly seen in the exposition of the results later.
DISCUSSION
The scientific discussion section lacks a comparative framework with other studies. References to other studies are missing. It would be interesting if the authors put more emphasis on justifying to what extent the results of the analysis corroborate or reject those obtained by previous studies (see e.g. doi: 10.1016/j.ijdrr.2020.101490). It would also be recommended to indicate to what extent the methodology used represents an advance in the disaster management field compared to existing alternative spatial analysis approaches for such phenomena (see e.g. doi: 10.6057/2018TCRR04.05) or other similar natural hazards (see e.g. doi:10.3390/app9153182).
CONCLUSION
The manuscript lacks a conclusions section summarizing the results obtained in the research. It is necessary to include this section so that readers can assess their interest in the results obtained at a simple glance.
Author Response

(The authors gave the same response as above.)

Round 2
Reviewer 1 Report
The manuscript is highly enhanced. I recommend to move the conclusion section to the end of the manuscript ( after discussion section)
Author Response
Comment: I recommend to move the conclusion section to the end of the manuscript ( after discussion section)
Response:Thank you for your suggestion. We have put the conclusion part after the discussion part.
Thanks again for your efforts so far for our manuscript.
Reviewer 3 Report
The authors have improved the manuscript, but I still think there is room for a more comprehensive literature review in the introduction. All the other requests were addressed in a good way.
Author Response
Comment: The authors have improved the manuscript, but I still think there is room for a more comprehensive literature review in the introduction.
Response:Thank you for your suggestion. We have minor revised the introduction of the manuscript (line 36-42, 101-102) to improve the quality of that part.
Thanks again for your efforts so far for our manuscript.
Reviewer 4 Report
All the concerns exposed in my previous report have been answered and the suggestions made to correct different shortcomings detected have been globally implemented in the new version of the manuscript. In my opinion the article can be considered for publication now.
Author Response
Thank you for your comment which are all valuable and very helpful for revising and improving our paper, as well as the important guiding significance to our researches.
Thanks again for your efforts so far for our manuscript.